# Normalizing Flows are Capable Generative Models

**Shuangfei Zhai** [1]   **Ruixiang Zhang** [1]   **Preetum Nakkiran** [1]   **David Berthelot** [1]   **Jiatao Gu** [1]   **Huangjie Zheng** [1]
**Tianrong Chen** [1]   **Miguel Angel Bautista** [1]   **Navdeep Jaitly** [1]   **Josh Susskind** [1]

## Abstract

Normalizing Flows (NFs) are likelihood-based models for continuous inputs. They have demonstrated promising results on both density estimation and generative modeling tasks, but have received relatively little attention in recent years. In this work, we demonstrate that NFs are more powerful than previously believed. We present TARFLOW: a simple and scalable architecture that enables highly performant NF models. TARFLOW can be thought of as a Transformer-based variant of Masked Autoregressive Flows (MAFs): it consists of a stack of autoregressive Transformer blocks on image patches, alternating the autoregression direction between layers. TARFLOW is straightforward to train end-to-end, and capable of directly modeling and generating pixels. We also propose three key techniques to improve sample quality: Gaussian noise augmentation during training, a post training denoising procedure, and an effective guidance method for both class-conditional and unconditional settings. Putting these together, TARFLOW sets new state-of-the-art results on likelihood estimation for images, beating the previous best methods by a large margin, and generates samples with quality and diversity comparable to diffusion models, for the first time with a stand-alone NF model. We make our code available at https://github.com/apple/ml-tarflow.

## 1. Introduction

Normalizing Flows (NFs) are a well-established likelihood based method for unsupervised learning (Tabak & Vanden-Eijnden, 2010; Rezende & Mohamed, 2015; Dinh et al., 2014). The method follows a simple learning objective,

[1]Apple. Correspondence to: Shuangfei Zhai <szhai@apple.com>.

*Proceedings of the $42^{nd}$ International Conference on Machine Learning*, Vancouver, Canada. PMLR 267, 2025. Copyright 2025 by the author(s).

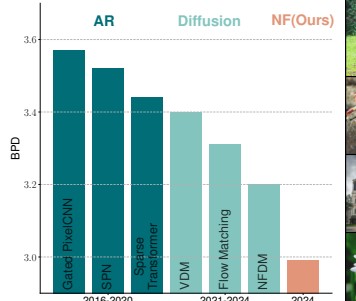 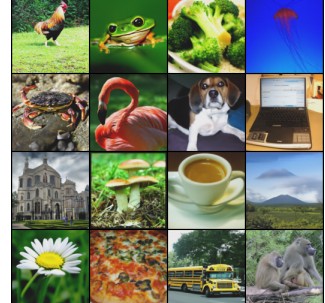

*Figure 1.* TARFLOW demonstrates substantial progress in the domain of normalizing flow models, achieving state-of-the-art results in both density estimation and sample generation. **Left**: We show the historical progression of likelihood performance on ImageNet 64x64, measured in bits per dimension (BPD), where our model significantly outperforms previous methods (see Table 2 for details). **Right**: Selected samples from our model trained on ImageNet 128x128 demonstrate unprecedented image quality and diversity for a normalizing flow model, establishing a new benchmark for this class of generative models.

which is to transform a data distribution into a simple prior distribution (such as Gaussian noise), keeping track of likelihoods via the change of variable formula. Normalizing Flows enjoy many unique and appealing properties, including exact likelihood computation, deterministic objective functions, and efficient computation of both the data generator and its inverse. There has been a large body of work dedicated to studying and improving NFs, and in fact NFs were the method of choice for density estimation for a number of years (Dinh et al., 2017; Kingma & Dhariwal, 2018; Chen et al., 2018; Papamakarios et al., 2017; Ho et al., 2019). However in spite of this rich line of work, Normalizing Flows have seen limited practical adoption— in stark contrast to other generative models such as Diffusion Models (Sohl-Dickstein et al., 2015; Ho et al., 2020) and Large Language Models (Brown et al., 2020). Moreover, the state-of-the-art in Normalizing Flows has not kept pace with the rapid progress of these other generative techniques, leading to less attention from the research community.

It is natural to wonder whether this situation is inherent – i.e., are Normalizing Flows fundamentally limited as a modeling paradigm? Or, have we just not found an appropriate

way to train powerful NFs and fully realize their potential? Answering this question may allow us to reopen an alternative path to powerful generative modeling, similar to how DDPM (Ho et al., 2020) enlightened the field of diffusion modeling and brought about its current renaissance.

In this work, we show that NFs are more powerful than previously believed, and in fact can compete with state-of-the-art generative models on images. Specifically, we introduce TARFLOW (short for Transformer AutoRegressive Flow): a powerful NF architecture that allows one to easily scale up the model's capacity; as well as a set of techniques that drastically improve the model's generation capability.

On the architecture side, TARFLOW is conceptually similar to Masked Autoregressive Flows (MAFs) (Papamakarios et al., 2017), where we compose a deep transformation by iteratively stacking multiple blocks of autoregressive transformations with alternating directions. The key difference is that we deploy a powerful masked Transformer (Vaswani et al., 2017) based implementation that operates in a block autoregression fashion (that is, predicting a block of dimensions at a time), instead of simple masked MLPs used in MAFs that factorizes the input on a per dimension basis.

In the context of image modeling, we implement each autoregressive flow transformation with a causal Vision Transformer (ViT) (Dosovitskiy et al., 2021) on top of a sequence of image patches, given a particular order of autoregression (e.g., top left to bottom right, or the reverse). This admits a powerful non-linear transformation among all image patches, while maintaining a parallel computational graph during training. Compared to other NF design choices (Dinh et al., 2017; Grathwohl et al., 2019; Kingma & Dhariwal, 2018; Ho et al., 2019) which often have several types of interleaving modules, our model features a modular design and enjoys greater simplicity, both conceptually and practically. This in return allows for much improved scalability and training stability, which is another critical aspect for high performance models. With this new architecture, we can immediately train much stronger NF models than previously reported, resulting in state-of-the-art results on image likelihood estimation.

On the generation side, we introduce three important techniques. First, we show that for perceptual quality, it is critical to add a moderate amount of Gaussian noise to the inputs, in contrast to a small amount of uniform noise commonly used in the literature. Second, we identify a post-training score based denoising technique that allows one to remove the noise portion of the generated samples. Third, we show for the first time that guidance (Ho & Salimans, 2022) is compatible with NF models, and we propose guidance recipes for both the class conditional and unconditional models. Putting these techniques together, we are able to achieve state-of-the-art sample quality for NF models on standard image modeling tasks.

We highlight our main results in Figure 1, and summarize our contributions as follows.

- We introduce TARFLOW, a simple and powerful Transformer based Normalizing Flow architecture.

- We achieve state-of-the-art results on likelihood estimation on images, achieving a sub-3 BPD on ImageNet 64x64 for the first time.

- We show that Gaussian noise augmentation during training plays a critical role in producing high quality samples.

- We present a post-training score-based denoising technique that allows one to remove the noise in the generated samples.

- We show that guidance is compatible with both class conditional and unconditional models, which drastically improves sampling quality.

*Table 1.* Notation.

| Notation | Meaning |
|----------|---------|
| $p_{\text{data}}(x)$ | training distribution |
| $p_{\text{model}}(y)$ | model distribution |
| $f(x)$ | the forward flow function |
| $f^t(z^t)$ | the forward function for the $t$-th flow block |
| $\mu^t, \alpha^t$ | learnable causal functions in the $t$-th flow block |
| $p_\epsilon(\epsilon)$ | the noise distribution |
| $q(y)$ | the noisy data distribution |
| $\tilde{p}(\tilde{x})$ | the discrete model distribution |

## 2. Method

### 2.1. Normalizing Flows

Given continuous inputs $x \sim p_{\text{data}}$, $x \in \mathbb{R}^D$, a Normalizing Flow learns a density $p_{\text{model}}$ via the change of variable formula $p_{\text{model}}(x) = p_0(f(x))|\det(\frac{df(x)}{dx})|$, where $f : \mathbb{R}^D \mapsto \mathbb{R}^D$ is an invertible transformation for which we can also compute the determinant of the Jacobian $\det(\frac{df(x)}{dx})$; $p_0$ is a prior distribution. The maximum likelihood estimation (MLE) objective can then be written as

$$\min_f \; -\log p_0(f(x)) - \log(|\det\left(\frac{df(x)}{dx}\right)|). \quad (1)$$

In this paper, we let $p_0$ be a standard Gaussian distribution $\mathcal{N}(0, I_D)$, so Equation 1 can be explicitly written as

$$\min_f \; 0.5\|f(x)\|_2^2 - \log(|\det\left(\frac{df(x)}{dx}\right)|), \quad (2)$$

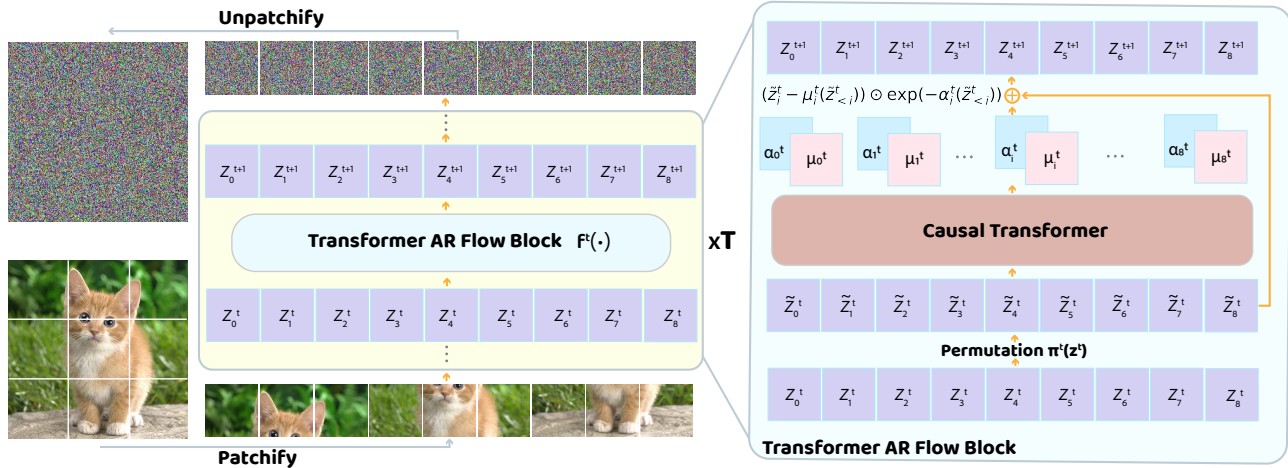

*Figure 2.* **Left**, TARFLOW consists of $T$ flow blocks trained end to end; **Right**, a zoom-in view of each flow bock, which contains a sequence permutation operation, a standard causal Transformer, and an affine transformation to the permuted inputs.

where we have omitted constant terms. Equation 2 bears an intuitive interpretation: the first term encourages the model to map data samples $x$ to latent variables $z = f(x)$ of small norm, while the second term discourages the model from "collapsing" — i.e., the model should map proximate inputs to separated latents which allows it to fully occupy the latent space. Once the model is trained, one automatically obtains a generative model via $z \sim p_0(z)$, $x = f^{-1}(z)$.

## 2.2. Block Autoregressive Flows

One appealing method for constructing a deep normalizing flow is by stacking multiple layers of autoregressive flows. This was first proposed in IAF (Kingma et al., 2016) in the context of variational inference, and later extended by MAF (Papamakarios et al., 2017) as standalone density models.

In this paper, we consider a generalized formulation of MAF — block autoregressive flows. Without loss of generality, we assume an input presented in the form of a sequence $x \in \mathbb{R}^{N \times D}$, where $N$ is the sequence length and $D$ is the dimension of each block of input. Let $T \in \mathbb{N}$ be the number of flow layers in the stack of flows. Subscripts denote indexing along the sequence dimension, e.g. $x_i \in \mathbb{R}^D$, and superscripts denotes flow-layer indices (see Figure 2). We then specify a flow transformation $z^T = f(x) := (f^{T-1} \circ f^{T-2} \cdots \circ f^0)(x)$ as follows. First, we choose $\{\pi^t\}$ as any fixed set of permutation functions along the sequence dimension. The $t$-th flow, $f^t$, is parameterized by two learnable functions $\mu^t, \alpha^t : \mathbb{R}^{N \times D} \to \mathbb{R}^{N \times D}$, which are both causal along the sequence dimension.

We initialize with $z^0 := x$. Then, the $t$-th flow transforms $z^t \in \mathbb{R}^{N \times D}$ into $z^{t+1} \in \mathbb{R}^{N \times D}$ by transforming a block of

inputs $\{z_i^t\}_{i \in [N]}$ as:

$$\tilde{z}^t \leftarrow \pi^t(z^t),$$

$$z_i^{t+1} \leftarrow \begin{cases} \tilde{z}_i^t & i = 0 \\ (\tilde{z}_i^t - \mu_i^t(\tilde{z}_{<i}^t)) \odot \exp(-\alpha_i^t(\tilde{z}_{<i}^t)) & i > 0 \end{cases} \quad (3)$$

Note that since $\mu^t$ is causal, the $i$-token of its output $\mu_i^t(\tilde{z}^t)$ only depends on $\tilde{z}_{<i}^t$, as written explicitly above. Iterating the above for $t = 0, 1, \ldots, (T-1)$ yields the output $z^T =: f(x)$. The inverse function $x = f^{-1}(z^T)$ is given by iterating the following flow to obtain $z^t$ from $z^{t+1}$:

$$\tilde{z}_i^t = \begin{cases} z_i^{t+1}, & i = 0 \\ z_i^{t+1} \odot \exp(\alpha_i^t(\tilde{z}_{<i}^t)) + \mu_i^t(\tilde{z}_{<i}^t) & i > 0 \end{cases}$$

$$z^t = (\pi^t)^{-1}(\tilde{z}^t). \quad (4)$$

This yields $x := z^0$ as the final iterate. As for the choice of permutations $\pi^t$, in this work we set all $\pi^t$ as the reverse function $\pi^t(z)_i = z_{N-1-i}$, except for $\pi^0$ which is set as identity. Ultimately, the entire flow transformation consists of $T$ flows $\{f^t\}$, and in each flow the input is first permuted then causally transformed with learnable element-wise subtractive and divisive terms $\mu_i^t(\cdot)$, $\exp(\alpha_i^t(\cdot))$.

It is worth noting that Equation 3 degenerates to MAF when $D = 1$. Intuitively, $D$ plays a role of balancing the difficulty of modeling each position in the sequence and the length of the entire sequence. This allows for extra modeling flexibility compared to the naive setting in MAF, which will become clearer in the later discussions.

In each flow transformation $f^t$, there are two operations. The first permutation operation $\pi^t$ is volume preserving, therefore its log determinant of the Jacobian is zero. The second autoregressive step has a Jacobian matrix of lower

triangular shape, which means its determinant needs to only account for the diagonal entries. The log determinant of the Jacobian then readily evaluates to

$$\log(|\det(\frac{df^t(z^t)}{dz^t})|) = -\sum_{i=1}^{N-1}\sum_{j=0}^{D-1}\alpha_i^t(\tilde{z}_{<i}^t)_j. \quad (5)$$

Putting them together, the training loss of our model can be written as

$$\min_f 0.5\|z^T\|_2^2 + \sum_{t=0}^{T-1}\sum_{i=1}^{N-1}\sum_{j=0}^{D-1}\alpha_i^t(\tilde{z}_{<i}^t)_j, \quad (6)$$

which simply consists of a square term and a sum of linear terms.

## 2.3. Transformer Autoregressive Flows

Architecture design is arguably the most challenging aspect of NF models. We suspect that a large part of the reason that NFs have not been as performant as other families of models is the lack of an architecture that allows for stable and scalable training.

To this end, we resort to a Transformer-based architecture, TARFLOW, with a design philosophy that features simplicity and modularity. In particular, we realize the fact that Equation 3 favors a parallel implementation with attention masks. This follows the same spirit as the original MAFs, but we replace the MLP based implementation with a much more powerful Transformer backbone which has a proven track record of success across both discrete and continuous domains. This seemingly simple change allows one to fully unlock the potentials of autoregressive flows, to a degree that has never been previously shown or expected.

We now consider the concrete case of modeling images, with the discussions generalizable to other domains. Given an image of shape $C \times H \times W$, where $C, H, W$ are the channel size, height and width of the image, respectively, we first convert it to a sequence of patches with a patch size of $S$. This gives us a sequence representation of $x \in \mathbb{R}^{N \times D}$, $N = \frac{HW}{S^2}$, $D = CS^2$. Similarly, the input of each flow transform $z^t$ will have the same size as $x$. We can then readily apply a standard Vision Transformer with causal attention masks to implement the transformation of a single autoregressive pass $f^t$. Importantly, the Transformer can have arbitrary depth and width, completely independent of the input's dimension.

When stacking multiple autoregressive transformations, the entire model can be viewed as a variant of a Residual Network. More specifically, the network consists of two types of residual connections: the first over the hidden layers inside the causal Transformer, the second over the latents $z_i^t$. This ensures another important factor of the architecture

design: training stability — i.e., training our model should be as easy as training a standard Transformer.

Combining the architecture and the loss (Equation 6) together, we have a complete recipe for a simple, scalable, and trainable NF model. See Figure 2 for an illustration of the architecture.

## 2.4. Noise Augmented Training

It is considered a common practice to introduce additive noise to the inputs during the training of NF models (Dinh et al., 2017; Ho et al., 2019). The usage of noise has mostly been motivated from the likelihood perspective, where adding uniform noise whose width is the same as the pixel quantization bin size to images allows one to "dequantize" the discrete pixel distribution to a continuous one. Formally speaking, instead of directly modeling the training data distribution $p_{\text{data}}$, we model a noise augmented distribution $q(y) = \int_\epsilon p_{\text{data}}(y - \epsilon)p_\epsilon(\epsilon)d\epsilon$. With a finite training set $\mathcal{X}$, this can be explicitly rewritten as $q(y) = \frac{1}{|\mathcal{X}|}\sum_{x\in\mathcal{X}} p_\epsilon(y - x)$. When evaluating likelihood, we follow the literature (Dinh et al., 2017) and let $p_\epsilon(\cdot) = \mathcal{U}(\cdot; 0, \text{bin})$, where bin is the quantization bin size (e.g., $\frac{1}{128}$ for 8-bit pixels normalized to the range of $[-1, 1]$). We can then compute likelihood w.r.t. the discrete inputs $\tilde{x}$ with $\tilde{p}(\tilde{x}) = \int_{\epsilon\in[0,\text{bin}]^D} p_{\text{model}}(\tilde{x} + \epsilon)d\epsilon$.

For better perceptual quality during sampling, however, we show that it is critical to set $p_\epsilon(\cdot)$ as a Gaussian distribution $\mathcal{N}(\cdot; 0, \sigma^2 \mathrm{I})$ whose magnitude $\sigma$ is small but larger than that of the pixel quantization bin size. To put it into context, with image pixels in $[-1, 1]$, an optimal $\sigma$ of $p_\epsilon(\cdot)$ for sample quality is around 0.05, whereas the standard deviation of the dequantization uniform noise is merely 0.002, an order of magnitude smaller.

Why is this the case? There are two factors which could be important. First, training a NF model with good generalization is inherently a challenging task. Without adding noise, the inverse model $f^{-1}(z)$ is effectively trained on discretized inputs $z$, of the same size as the training set. During the inference, however, $f^{-1}$ is expected to generalize on a much denser input distribution (e.g., Gaussian), which poses an out-of-distribution problem that hinders the sampling quality. Adding noise therefore serves a simple purpose of enriching the support of the training distribution, hence the support of the inverse model $f^{-1}$. Second, using a Gaussian noise instead of uniform is also critical, as the former effectively stretches the support of the training distribution to the ambient input space, with the mode of the density placed at the original data points. Although this makes it less straightforward to convert the learned density $q(y)$ to a discrete data probability, but we will later see that it greatly enhances the sampling quality.

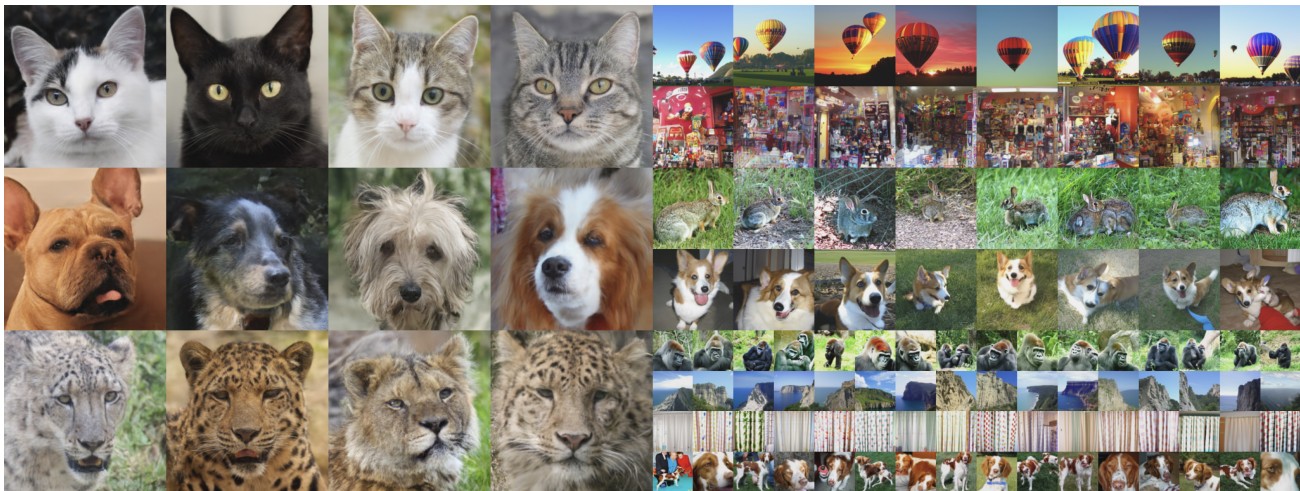

*Figure 3.* Images of various resolutions generated by TARFLOW models. From left to right, top to bottom: 256x256 images on AFHQ, 128x128 and 64x64 images on ImageNet.

### 2.5. Score Based Denoising

Training with noise augmentation introduces an additional challenge: models trained on the noisy distribution $q(y)$ naturally generate outputs that mimic noisy training examples, rather than clean ones. This results in samples that are less visually appealing. As a remedy, we propose a straightforward training-free technique that effectively denoises the generated samples, by drawing inspiration from score-based generative models.

The idea is as follows. Consider the joint distribution $(x, y)$ where $x \sim p_{\text{data}}$ and $y = x + \varepsilon$ for $\varepsilon \sim \mathcal{N}(0, \sigma^2 \mathrm{I})$. By definition, $y$ is marginally distributed as the noisy data distribution $q$. By Tweedie's formula, we have

$$\mathbb{E}[x \mid y] = y + \sigma^2 \nabla_y \log q(y). \tag{7}$$

Therefore, given a noisy sample $y$ we can denoise it to a clean sample $\hat{x} := \mathbb{E}[x \mid y]$ if we know gradients of the log-likelihood $\log q(y)$. Under the condition when $\sigma$ is small, we have $\mathbb{E}[x \mid y] \approx x$. Now further assuming that the model $p_{\text{model}}(\cdot)$ is well trained, we can use the same formula to denoise a sample from the model, using $p_{\text{model}}$ in place of $q$. The complete sampling procedure can be written as:

$$z \sim p_0, y := f^{-1}(z), x := y + \sigma^2 \nabla_y \log p_{\text{model}}(y). \tag{8}$$

It is worth noting that our score based denoising technique uses only the TARFLOW model itself, without requiring any extra modules. The fact that this works well (as demonstrated later in Sec. 3.3) suggests that learning the density of the noisy distribution is sufficient for recovering the score function, which is an interesting and significant result on its own in the context of score based generative models.

### 2.6. Guidance

An important property of state-of-the-art generative models is their ability to be controlled during inference. Normalizing flows have conventionally relied on low temperature sampling (Kingma & Dhariwal, 2018), but it's only applicable to the volume preserving variants and also introduces severe smoothing artifacts.

On the other hand, guidance in diffusion models (Dhariwal & Nichol, 2021; Ho & Salimans, 2022) have achieved great success in this regard, which allows one to trade-off diversity for improved mode seeking ability. Surprisingly, we found that our models can also be guided, offering very similar flexibility to the case in diffusion models.

In the conditional generation setting, guidance can be obtained in almost the exact same way as classifier free guidance (CFG) (Ho & Salimans, 2022) in diffusion models. We first override the notation by letting $\mu_i^t(\cdot; c), \alpha_i^t(\cdot; c)$ be the class conditional predictions, and $\mu_i^t(\cdot; \emptyset), \alpha_i^t(\cdot; \emptyset)$ be the unconditional counterparts. In practice, the unconditional predictions can be obtained by randomly dropping out the class label during training, similar to (Ho & Salimans, 2022). For each flow block $t$, we modify the reverse function in Equation 4 to

$$\tilde{z}_i^t = z_i^{t+1} \odot \exp(\tilde{\alpha}_i^t(\tilde{z}_{<i}^t; c, w)) + \tilde{\mu}_i^t(\tilde{z}_{<i}^t; c, w). \tag{9}$$

Here we generate $\tilde{z}_i^t$ with the guided predictions $\tilde{\mu}_i^t(\cdot; c, w), \tilde{\alpha}_i^t(\cdot; c, w)$ under guidance weight $w$, which are defined as

$$\begin{aligned}
\tilde{\mu}_i^t(\tilde{z}_{<i}^t; c, w) &= (1+w)\mu_i^t(\tilde{z}_{<i}^t; c) - w\mu_i^t(\tilde{z}_{<i}^t; \emptyset), \\
\tilde{\alpha}_i^t(\tilde{z}_{<i}^t; c, w) &= (1+w)\alpha_i^t(\tilde{z}_{<i}^t; c) - w\alpha_i^t(\tilde{z}_{<i}^t; \emptyset).
\end{aligned} \tag{10}$$

Intuitively, under positive guidance $w > 0$, Equation 10

modifies the updates of sampling to guide conditional variables $\tilde{z}_i^t$ away from predictions from an unconditional model, therefore converging more towards the class model of $c$.

We have also discovered as method for applying guidance to unconditional models, see Appendix Section B for details.

## 3. Experiments

We perform our experiments on unconditional ImageNet 64x64 (van den Oord et al., 2016b), as well as class conditional ImageNet 64x64, ImageNet 128x128 (Deng et al., 2009) and AFHQ 256x256 (Choi et al., 2020).

Our models are implemented as stacks of standard causal Vision Transformers (Dosovitskiy et al., 2021). In each AR flow block, the inputs are first linearly projected to the model channel size, then added with learned position embeddings. For class conditional models, we add an immediate class embedding on top of it. We use attention head dimensions of $64$ and an MLP latent size $4\times$ that of the model channel size. The output layer of each flow block consists of two heads per position, corresponding to $\mu_i^t, \alpha_i^t$, respectively, and they are initialized as zeros. All parameters are trained end-to-end with the AdamW optimizer with momentum $(0.9, 0.95)$. We use a cosine learning rate schedule, where the learning rate is warmed up from $10^{-6}$ to $10^{-4}$ for one epoch, then decayed to $10^{-6}$. We use a small weight decay of $10^{-4}$ to stabilize training.

We adopt a simple data preprocessing protocol, where we center crop images and linearly rescale the pixels to $[-1, 1]$. For each task, we search for architecture configurations consisting of the patch size (P), model channel size (Ch), number of autoregressive flow blocks (T) and the number of attention layers in each flow (K). For generation tasks, we also search for the best input noise $\sigma$ that yields the best sampling quality. We denote a TARFLOW configuration as P-Ch-T-K-$p_\epsilon$.

### 3.1. Likelihood

Likelihood estimation provides a direct assessment of a normalizing flow architecture's modeling capacity, as it aligns precisely with the model's training objective. For evaluating likelihood on image data, unconditional ImageNet 64x64 has acted as the de facto benchmark dataset. Its relatively large scale and inherent diversity pose significant challenges for model fitting, making it an ideal testbed where improvements typically stem from enhanced model capacity rather than regularization techniques.

During both training and evaluation, we apply uniform noise $\mathcal{U}(0, \frac{1}{128})$ to the data, which corresponds to the "dequantization" noise (Dinh et al., 2017). We do not use any additional

*Table 2.* Bits per dim evaluation on unconditional ImageNet 64x64 test set. We denote the TARFLOW configuration in the format [P-Ch-T-K-$p_\epsilon$].

| Model | Type | BPD↓ |
|---|---|---|
| Very Deep VAE (Child, 2021) | VAE | 3.52 |
| Glow (Kingma & Dhariwal, 2018) | Flow | 3.81 |
| Flow++ (Ho et al., 2019) | Flow | 3.69 |
| PixelCNN (van den Oord et al., 2016a) | AR | 3.83 |
| SPN (Menick & Kalchbrenner, 2019) | AR | 3.52 |
| Sparse Transformer (Child et al., 2019) | AR | 3.44 |
| Routing Transformer (Roy et al., 2021) | AR | 3.43 |
| Improved DDPM (Nichol & Dhariwal, 2021) | Diff/FM | 3.54 |
| VDM (Kingma et al., 2021) | Diff/FM | 3.40 |
| Flow Matching (Lipman et al., 2023a) | Diff/FM | 3.31 |
| NFDM (Bartosh et al., 2024) | Diff/FM | 3.20 |
| TARFLOW [2-768-8-8-$\mathcal{U}(0, \frac{1}{128})$] (Ours) | NF | **2.99** |

*Table 3.* Fréchet Inception Distance (FID) evaluation on Conditional ImageNet $64\times64$. We denote the TARFLOW configuration in the format [P-Ch-T-K-$p_\epsilon$].

| Model | Type | FID↓ |
|---|---|---|
| EDM (Karras et al., 2022) | Diff/FM | 1.55 |
| iDDPM (Nichol & Dhariwal, 2021) | Diff/FM | 2.92 |
| ADM(dropout) (Dhariwal & Nichol, 2021) | Diff/FM | 2.09 |
| IC-GAN (Casanova et al., 2021) | GAN | 6.70 |
| BigGAN (Brock et al., 2019) | GAN | 4.06 |
| CD(LPIPS)(Song et al., 2023) | CM | 4.70 |
| iCT-deep(Song & Dhariwal, 2023) | CM | 3.25 |
| TARFLOW [4-1024-8-8-$\mathcal{N}(0, 0.05^2)$] (Ours) | NF | 3.99 |
| TARFLOW [2-768-8-8-$\mathcal{N}(0, 0.05^2)$] (Ours) | NF | 2.90 |
| TARFLOW [2-1024-8-8-$\mathcal{N}(0, 0.05^2)$] (Ours) | NF | 2.66 |

*Table 4.* Fréchet Inception Distance (FID) evaluation on Conditional ImageNet $128\times128$. We denote the TARFLOW configuration in the format [P-Ch-T-K-$p_\epsilon$].

| Model | Type | FID↓ |
|---|---|---|
| ADM-G (Dhariwal & Nichol, 2021) | Diff/FM | 2.97 |
| CDM (Ho et al., 2022) | Diff/FM | 3.52 |
| Simple Diff (Hoogeboom et al., 2023) | Diff/FM | 1.94 |
| RIN (Jabri et al., 2023) | Diff/FM | 2.75 |
| BigGAN (Brock et al., 2019) | GAN | 8.70 |
| BigGAN-deep (Brock et al., 2019) | GAN | 5.70 |
| TARFLOW [4-1024-8-8-$\mathcal{N}(0, 0.05^2)$] (Ours) | NF | 5.29 |
| TARFLOW [4-1024-8-8-$\mathcal{N}(0, 0.15^2)$] (Ours) | NF | 5.03 |

data augmentation techniques during training. As shown in Table 2 and visualized in Figure 1, our approach establishes new state-of-the-art result in test set likelihood, by a significant margin over all previous models.

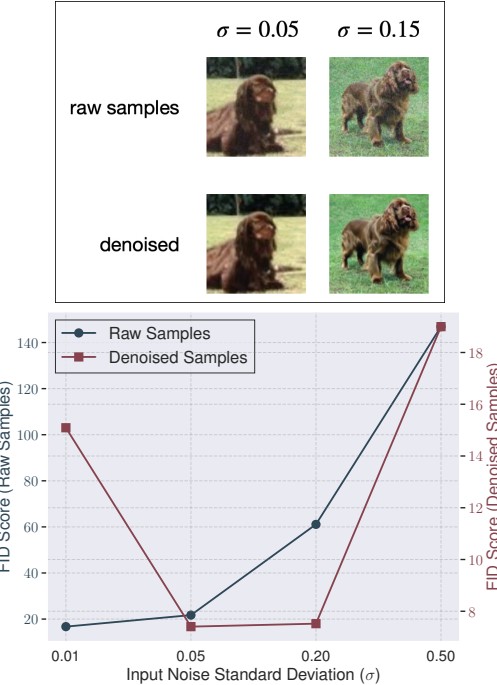

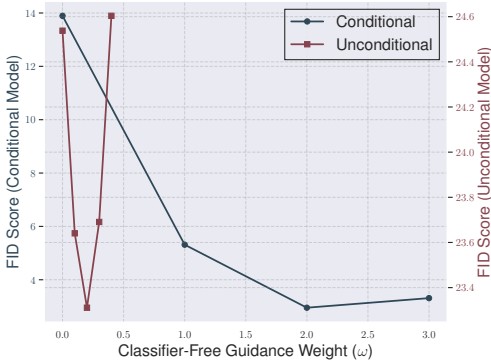

*Figure 5.* Guidance weight $w$ vs FID for both the conditional and unconditional models (with $\tau = 1.5$) on ImageNet 64x64. Note the y axis's scale difference between the two settings.

*Figure 4.* **Top**: The effect of input noise $\sigma$ and denoising, all samples are generated with guidance weight $w = 2$ on ImageNet 128x128 from the same initial noise, better viewed when zoomed in. **Bottom**: Sample FID vs input noise $\sigma$ on ImageNet 64x64, with and without denoising. Before denosing, it first appears that small $\sigma$ has the best FID, due to the smaller amount of noise present in the raw samples. However, after denoising with Equation 8, slightly larger $\sigma$ favors better FID and demonstrates more consistent shapes. Note that the *scale* of the right y-axis differs from that of the left.

### 3.2. Generation

Next, we evaluate TARFLOW's sampling ability in class conditional (ImageNet 64x64, ImageNet 128x128, AFHQ 256x256) as well as unconditional (ImageNet 64x64) settings. Our experimental protocol is largely the same as previously mentioned, except that we adopt random horizontal image flips. For the class conditional models, we randomly drop the class label with a probability of 0.1.

We first show qualitative results in Figure 3, which are obtained with the sampling procedure in Equation 8. We see that TARFLOW generates diverse and high fidelity images in all settings. Also, TARFLOW seems to demonstrate great robustness w.r.t. the data size and resolution. For instance, it works well on both a large diverse dataset (ImageNet, $\sim 1.3M$ examples, 1K classes) and a small but high resolution one (AFHQ, $15K$ examples in 3 classes and 256x256). Visually, these samples are comparable to those generated by Diffusion Models, which marks a large improvement from the previous best NF models. We include more qualitative results in the Appendix.

We then perform quantitative evaluations in terms of FID, on the ImageNet models. For each setting, we randomly generate 50K samples, and compare it with the statistics from the entire training set. We search for the best guidance weights (and attention temperature in the unconditional case). The results are summarized in Table 3, 6, 4. In all settings, we see that TARFLOW produces competitive FID numbers, often times better than strong GAN baselines, and approaching results from recent Diffusion Models. It is also interesting to note that we found no publicly reported NF based FID numbers on the ImageNet level datasets, most likely due to the lack of presentable results from the NF community.

### 3.3. Ablation on Noise Augmentation and Denoising

We then study the role of input noise $p_\epsilon$. We first experimented with the 'dequantization' uniform noise and found that sampling experiences constant numerical issues and was not able to produce sensible outputs. We hypothesize the reason being that a narrow uniform noise makes the flow transformation ill-conditioned, as it forces a model to map a low entropy distribution to an ambient Gaussian distribution.

Next, we experiment with different Gaussian noise levels $\sigma$ during training on class conditional ImageNet 64x64. We use an architecture configuration of 4-1024-8-8, and vary $\sigma$ in $\{0.01, 0.05, 0.2, 0.5\}$. For fast experimentation, we train all models for only 100 epochs with a batch size of 512. We evaluate these models with a guidance $w = 2$ and plot the 50K sample FIDs before and after the score based denoising. For visual comparison, we also train two models ImageNet 128x128 models, with the architecture 4-1024-8-8 and noise $\sigma \in [0.05, 0.15]$, respectively. We show the FID curves together with the visual examples in Figure 4. There are two important observations. First, naively increasing the noise level on the surface appears to

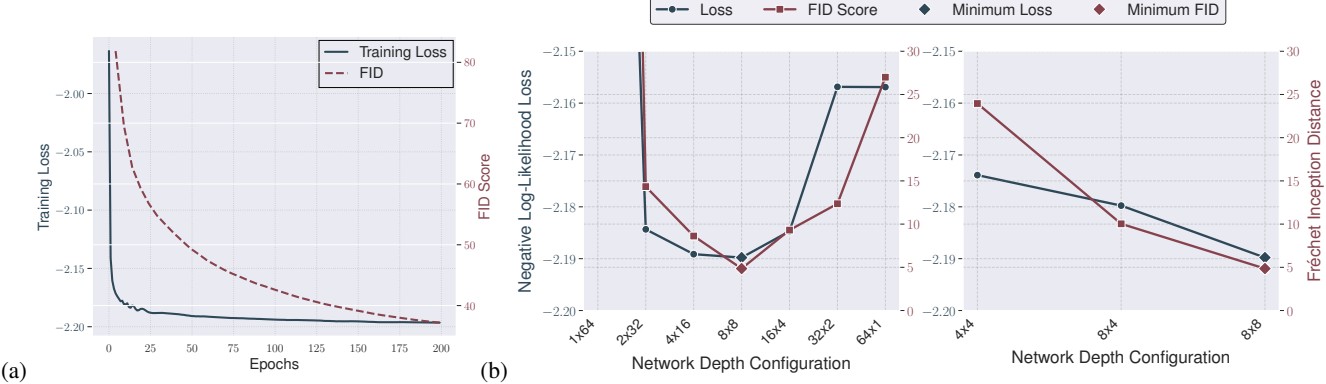

*Figure 6.* (a) A typical training run on ImageNet 64x64. Our loss smoothly decreases during training, and is positively correlated with FID. (b) Depth configuration (in the form of $T \times K$) vs training loss. Overall, we see a strong positive correlation between the training loss and FID. **Left**, the optimal training loss happens when the capacity is evenly allocated to number of blocks and number of layers per block. Interestingly, the special case of 1 block degenerates to an incapable model, which has both high loss and FID equivalent to random guess (FID 267). **Right**, increasing both the number of blocks and number of layers per block improves the model's loss and sampling quality.

hurt the raw samples' quality. However, this is no longer the case after applying the denoising step in Equation 8. Denoising successfully cleans up the noisy raw samples, and as a result the best visual quality occurs at a moderate (but still relatively small) amount of noise. This verifies the necessity of our proposed sampling procedure, whereas the combination of noise augmented training and the score based denoising step work organically together to produce the best generative capability. See the Appendix for more visualizations of the effect of denoising.

### 3.4. Ablation on Guidance

To see the effect of guidance, we perform qualitative and quantitative evaluations on both the class conditional and unconditional versions of ImageNet 64x64. The results are shown in Figure 5 and 7 (in Appendix). In terms of FID, the guidance weight $w$ plays an effective role for both models. Visually, it is also clear that guidance allows the model to converge to more recognizable modes, presenting more aesthetic samples. Interestingly, this is also somewhat true for the unconditional models, whereas both the guidance weight $w$ and attention temperature $\tau$ contribute to the degree of guidance. We show more guidance comparisons in the Appendix.

### 3.5. Ablation on Model Scaling

Regarding scaling, we first show a typical training loss curve together with an online monitoring of the model's sample quality in terms of FID (we use 4096 samples for efficiency). This is shown in Figure 6(a). We see that the loss curve is smooth and monotonic, and it has a strong positive correlation with the FID curve.

We proceed to discuss another design question: the model's

size, especially model's depth. Depth plays a vital role in our model, as we need to have a sufficient number of flow blocks, as well as number of layers within each block. This deep transformation then poses questions on architecture design as well as its trainability.

We answer this question by performing two sets of ablations on conditional ImageNet 64x64. In the first set of experiments, we train a set of models who share the same number of combined layers $T \times K$; and in the second, we increase a base model's depth by increasing either $T$ or $K$. The results are shown in Figure 6(b). First of all, we observe again the strong positive correlation between the loss and FID curves, across different architectures. This points to a nice property of NF models where improving the likelihood (i.e., the loss) directly leads to improved generative modeling capabilities. Second, there is a U-shape distribution w.r.t. the $T \times K$ configuration, and it appears that the best trade-off occurs when $T = K$. The case of $T = 1$ is also interesting, as it corresponds to a special case of a single direction autoregressive model on image patches. It is obvious that this model fails to fit the data, both in terms of loss and FID. This is in contrast to the $T = 2$ configuration which has a much more reasonable performance. Lastly, increasing either $T$ or $K$ is effective in improving the model's capacity. Putting these observations together, we see that TARFLOW demonstrates promising scaling behaviors, which makes it a particularly appealing candidate for exploiting the wide abundance of power of modern compute infrastructures.

### 3.6. Comparison with VP and Channel Coupling

We also ablation two important design choices of TARFLOW: the non-volume preserving (NVP) and autoregressive aspect. We train two baseline models: one to change NVP to VP

*Table 5.* VP and channel coupling w.r.t. FID on ImageNet 64×64.

| Guidance | TARFLOW | VP | channel coupling |
|----------|---------|------|------------------|
| 0 | 25.3 | 81.5 | 50.3 |
| 2 | 5.7 | 51.0 | 20.4 |

(done by setting $\alpha_i^t(\cdot)$ to 0 in Equation 3); and the other a channel coupling baseline by using the same architecture but removing the causal masks. We use the same 0.05 Gaussian noise on both, and found that and denoising consistently help. For the VP baseline, we found that we also need to learn the prior's variance as the latents tend to have very small magnitudes which makes sampling from the standard Gaussian prior immediately fail. All models share similar training costs which makes it a fair comparison. The results are shown in Table 5. We see that both variants significantly under perform TARFLOW. Also, interestingly, guidance consistently improves both settings.

## 4. Related Work

**Coupling-based Normalizing Flows.** NICE (Dinh et al., 2014) introduced additive coupling layers to construct the transformations and simplified the computation of the Jacobian determinant. RealNVP (Dinh et al., 2017) extended this approach by incorporating scaling and shifting operations to enhance the model's expressiveness. Glow (Kingma & Dhariwal, 2018) advanced these models by introducing invertible $1 \times 1$ convolutions, achieving improved results in image generation tasks. Flow++ (Ho et al., 2019) further introduced learned dequantization noise, a sophisticated coupling layer and attention mechanisms to enhance the model's expressiveness. What's shared in common among these designs is that they need carefully wired and restrictive architectures, which poses great a challenge in scaling the model's capacity.

**Continuous Normalizing Flows.** Neural Ordinary Differential Equations (Chen et al., 2018) based Continuous Normalizing Flows is an alternative NF design principle. In this framework, the invertibility of the network is inherently satisfied, and the computation of the Jacobian determinant within the normalizing flow is reduced to calculating the trace of the Jacobian. FFJORD (Grathwohl et al., 2019) further simplifies the expensive Jacobian computation by employing Hutchinson's trace estimator (Hutchinson, 1989). However, these models often suffer from numerical instability during training and sampling, which has been extensively analyzed in (Zhuang et al., 2021; Liu et al., 2021). The expressive capability can be further improved by augmenting auxiliary variables (Dupont et al., 2019; Chalvidal et al., 2021). In comparison, TARFLOW enables an unconstrained architecture design paradigm by fully taking advantage of the power of causal Transformers, which we believe is a key component for realizing the true potential of the NF principle.

**Autoregressive Normalizing Flows.** IAF (Kingma et al., 2016) introduced dimension-wise affine transformations conditioned on preceding dimensions for variational inference, and MAF (Papamakarios et al., 2017) leveraged the MADE (Germain et al., 2015) architecture to construct invertible mappings through autoregressive transformations. Neural autoregressive flow (Huang et al., 2018) replaces the affine transformation in MAF by parameterizing a monotonic neural network, at the cost of losing analytical invertibility. T-NAF (Patacchiola et al., 2024) extends NAF by introducing a single autoregressive Transformer. Block Neural Autoregressive Flow (Cao et al., 2019) fits an end-to-end autoregressive monotonic neural network, rather than NAF's dimension-wise sequence parameterization, but also sacrifices analytical invertibility. TARFLOW differs from these as we show that it is sufficient to stack multiple iterations of block autoregressive flows with standard Transformer model in alternating directions, without the need for other types of flow operations.

**Probability Flow in Diffusion & Flow Matching.** Diffusion models (Sohl-Dickstein et al., 2015; Ho et al., 2020; Song et al., 2021) generate data by simulating Stochastic Differential Equations. Song et al. (2021) provided a deterministic Ordinary Differential Equation (ODE) counterpart to this generative approach, also known as the probability flow (Song et al., 2021). Similarly, Flow Matching (Lipman et al., 2023a) proposes to learn such ODEs by training on linear interpolants of data and noise with the velocity prediction objective. Importantly, TARFLOW differs from these instances as it is directly trained with the MLE objective, without the need for excessively large Gaussian noise during training.

See also Sec. A in Appendix for additional related works.

## 5. Conclusion

We presented TARFLOW, a Transformer-based architecture together with a set of techniques that allows us to train high-performance normalizing flow models. Our model achieves state-of-the-art results on likelihood estimation, improving upon the previous best results by a large margin. We also show competitive sampling performance, qualitatively and quantitatively, and demonstrate for the first time that normalizing flows alone are a capable generative modeling technique. We hope that our work can inspire future interest in further pushing the envelope of simple and scalable generative modeling principles.

## Acknowledgements

We thank Yizhe Zhang, Alaa El-Nouby, Arwen Bradley, Yuyang Wang, and Laurent Dinh for helpful discussions. We also thank Samy Bengio for leadership support that made this work possible.

## Impact Statement

This paper concerns the generative modeling methodology. While we do not see immediate societal implications from our technical contribution, there are potential impacts when it is used in training foundational generative models.

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

## A. Additional Related Work

**Autoregressive Models for Image Generation**   Many efforts (van den Oord et al., 2016b;a; Esser et al., 2021; Razavi et al., 2019) have been made to apply autoregressive sequential methods to image generation. PixelRNN(van den Oord et al., 2016b) is a pioneering work in this field. This approach views an image as a sequence of data, modeling the distribution of each subsequent pixel conditioned on all previously generated pixels through an RNN architecture(Sherstinsky, 2020). This methodology is readily adaptable to masked convolutional structures(van den Oord et al., 2016a), where the prediction of the next pixel is based on its neighboring pixels, bypassing the use of a traditional convolutional kernel. The transformer model has been successfully applied to image generation tasks. Chen et al. (2020) introduced ImageGPT, an autoregressive model that predicts pixels sequentially in raster order. More recently, Yu et al. (2022) introduced Parti, a scalable encoder-decoder transformer for text-to-image generation, which conceptualizes the task as a sequence-to-sequence problem. VAR(Tian et al., 2024) begins with low-resolution images in the latent space, effectively predicting the next level of resolution and yielding impressive outcomes. Studies like (Yu et al., 2022; Gu et al., 2024; Sun et al., 2024) further demonstrate the scalability and effectiveness of autoregressive models in producing high-dimensional images. Recent work MAR (Li et al., 2024) introduced diffusion models for autoregressive latent token prediction as an alternative to vector quantization approaches for image generation. GIVT (Tschannen et al., 2025) employed transformer decoders to model latent tokens generated by a VAE encoder, while incorporating a Gaussian Mixture Model (GMM) in place of categorical prediction for likelihood modeling. Concurrently to our work, JetFormer (Tschannen et al., 2024) further extended the approach by substituting the VAE with a coupling-based normalizing flow model, and used an autoregressive Transformer with GIVT's GMM prediction head to model sequences of latent tokens. While TARFLOW employs a causal Transformer architecture similar to these approaches, it operates differently by processing continuous data directly with a single model, thus avoiding the complexity of input discretization, or the need for separate image tokenization and autoregressive modeling stages.

**Diffusion models, other generative models**   Diffusion models (Ho et al., 2020; Song et al., 2021) are emerging generative models that achieve appealing results. Stable Diffusion (Podell et al., 2024) and OpenSora (Zheng et al., 2024) push the boundaries of diffusion models' capabilities, demonstrating their ability to generate extremely high-dimensional data. Besides, Variational Autoencoders (VAEs) (Kingma & Welling, 2014) and Generative Adversarial Networks (GANs) (Goodfellow et al., 2014) are also popular generative models. By avoiding the posterior collapse issue, VQ-VAE (van den Oord et al., 2017) demonstrates impressive generative performance and subsequently serves as an essential component in the later latent diffusion model (Podell et al., 2024). In the realm of GANs, Karras et al. (2019); Kang et al. (2023); Brock et al. (2019) showcase the remarkable capability of GANs to generate high-resolution images with comparatively cheap inference costs, though the training stability of GANs remains challenging (Wiatrak et al., 2019). TARFLOW represents an orthogonal learning paradigm to these methods, with its unique benefits and challenges.

## B. Guidance

In addition to conditional guidance, we also introduce a novel method for guiding unconditional models. The basic idea is to construct predictions of inferior quality, analogous to the role of unconditional ones. In order to do so, we override the notation yet again and introduce $\mu_i^t(\cdot;\tau), \alpha_i^t(\cdot;\tau)$. Here we have let the predictions $\mu_i^t, \alpha_i^t$ take an additional parameter $\tau$, which corresponds to a manually injected temperature term to all the attention layers in the Transformer for $f^t$. Namely, for each attention layer in $f^t$, we divide the attention logits by $\tau$, before normalizing it with the Softmax function. A $\tau$ larger or smaller than 1 makes the attention overly smooth or sharp, either way reducing the Transformer's ability to correctly predict the next variable's transformations.

We can then similarly write out the unconditional guided predictions as

$$\tilde{\mu}_i^t(\tilde{z}_{<i}^t;\tau,w) = (1+w)\mu_i^t(\tilde{z}_{<i}^t;1) - w\mu_i^t(\tilde{z}_{<i}^t;\tau),$$
$$\tilde{\alpha}_i^t(\tilde{z}_{<i}^t;\tau,w) = (1+w)\alpha_i^t(\tilde{z}_{<i}^t;1) - w\alpha_i^t(\tilde{z}_{<i}^t;\tau),$$
(11)

where increasing either $w$ or $|\tau - 1|$ corresponds to stronger guidance.

Lastly, for both the conditional and unconditional cases, it is possible to assign a different guidance weight $w_i^t$ depending on the flow and position index $t, i$. We have preliminarily explored a linearly increased $w_i$ as a function of $i$, as in $w_i = \frac{i}{T-1}w$, and we have found this to achieve better sampling results w.r.t. FID than uniform guidance weights. We leave the thorough exploration of the optimal guidance schedule as future work.

*Table 6.* Fréchet Inception Distance (FID) evaluation on Unonditional ImageNet 64×64. We denote the TARFLOW configuration in the format [P-Ch-T-K-$p_\epsilon$].

| Model | Type | FID $\downarrow$ |
|---|---|---|
| MFM (Pooladian et al., 2023) | Diff/FM | 11.82 |
| FM (Lipman et al., 2023b) | Diff/FM | 13.93 |
| AGM (Chen et al., 2024) | Diff/FM | 10.07 |
| IC-GAN (Casanova et al., 2021) | GAN | 10.40 |
| Self-sup GAN (Noroozi, 2020) | GAN | 19.20 |
| TARFLOW [2-768-8-8-$\mathcal{N}(0, 0.05^2)$] (Ours) | NF | 18.42 |

## C. Experimental details

Our models are implemented with PyTorch, and our experiments are conducted on A100 GPUs. We by default cast the model to *bfloat16*, which provides significant memory savings, with the exception of the likelihood task where we found that *float32* is necessary to avoid numerical issues. All of our jobs are finished within 14 days of training, though we believe that the models should get better if trained longer. We summarize the hyperparameters for our best jobs in Table 7.

*Table 7.* Hyper parameters for the best performing model on each task.

| Task | Patch Size | Channels | Num Flows | Layers per Flow | Input Noise | Batch size | Epochs | Num GPUs |
|---|---|---|---|---|---|---|---|---|
| Uncond ImageNet 64x64 (likelihood) | 2 | 768 | 8 | 8 | $\mathcal{U}(0, \frac{1}{128})$ | 384 | 60 | 32 |
| Uncond ImageNet 64x64 (generation) | 2 | 768 | 8 | 8 | $\mathcal{N}(0, 0.05^2)$ | 256 | 200 | 8 |
| Cond ImageNet 64x64 | 2 | 1024 | 8 | 8 | $\mathcal{N}(0, 0.05^2)$ | 768 | 320 | 32 |
| Cond ImageNet 128x128 | 4 | 1024 | 8 | 8 | $\mathcal{N}(0, 0.15^2)$ | 768 | 320 | 32 |
| Cond AFHQ 256x256 | 8 | 768 | 8 | 8 | $\mathcal{N}(0, 0.07^2)$ | 256 | 4000 | 8 |

## D. Inference Implementation

Although our main focus in this paper has been on training capable generative models, it is still worth commenting on the sampling efficiency of our method. Sampling from a TARFLOW involves reversing a series of causal Transformers. Unlike the training model where the autoregressive flow can be computed in parallel with causal masks, the reverse step is inevitably sequential with respect to the sequence direction. In practice, we resort to a KV-cache based implementation, which is a standard practice in the context of LLMs, and we found that it greatly speeds up the sampling over a naive implementation. For instance, sampling from a guided batch of 32 samples from the 2-768-8-8-$\mathcal{N}(0, 0.05^2)$ ImageNet 64x64 model takes about 2 minutes on a single A100 GPU. Although efficient sampling is not the focus of this work, we believe that there is great room for improvement in this regard, and we leave it as future work.

Another component in our sampling pipeline is the score based denoising step. The time of this step is equal to two forward model calls, which usually happens in a matter of seconds. A practical bottleneck is that this step is more memory consuming than the flow reverse step, due to the need of caching all intermediate activations for back propagation. In theory, this can be further alleviated by adopting techniques like gradient checkpointing, essentially trading time for memory.

### D.1. Visualizing Sample Trajectory

Thanks to the residual style composition of TARFLOW, we can also visualize the generation process by reshaping each $\{z^t\}$ to the pixel space. We visualize two sampling sequences with the ImageNet 128x128 model in Figure 8. Interestingly, the sample trajectories highly resemble those from a Diffusion model, in the sense that the initial noise is gradually transformed into visible inputs – though they are trained with completely different objectives.

## E. Additional samples

Next we show more uncurated samples from four generation tasks, demonstrating the raw samples, guided samples as well as denoised samples in Figure 9, 10, 11 and 12.

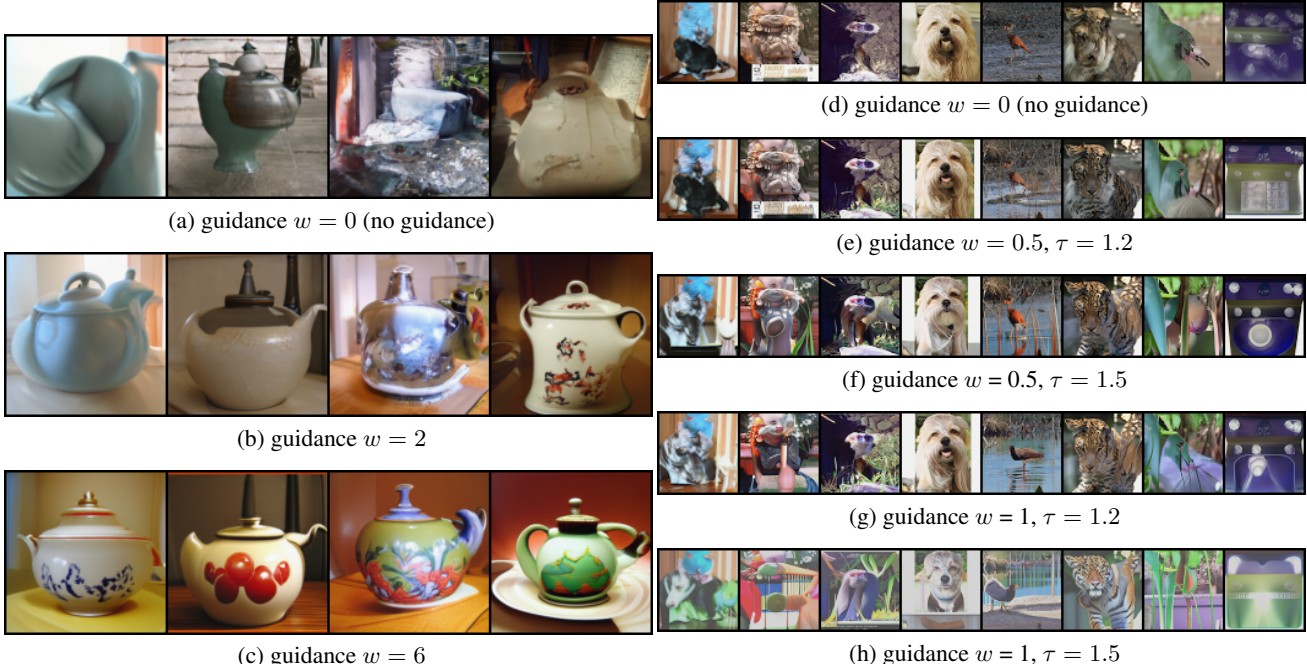

(a) guidance $w = 0$ (no guidance)

(b) guidance $w = 2$

(c) guidance $w = 6$

(d) guidance $w = 0$ (no guidance)

(e) guidance $w = 0.5, \tau = 1.2$

(f) guidance $w = 0.5, \tau = 1.5$

(g) guidance $w = 1, \tau = 1.2$

(h) guidance $w = 1, \tau = 1.5$

*Figure 7.* **Left**:Varying guidance weight with the class conditional model on ImageNet 128x128, here we show 4 samples from the ImageNet class 849 ("teapot"); **Right**: Varying guidance weight and attention temperature for the uncondtional ImageNet 64x64 model.

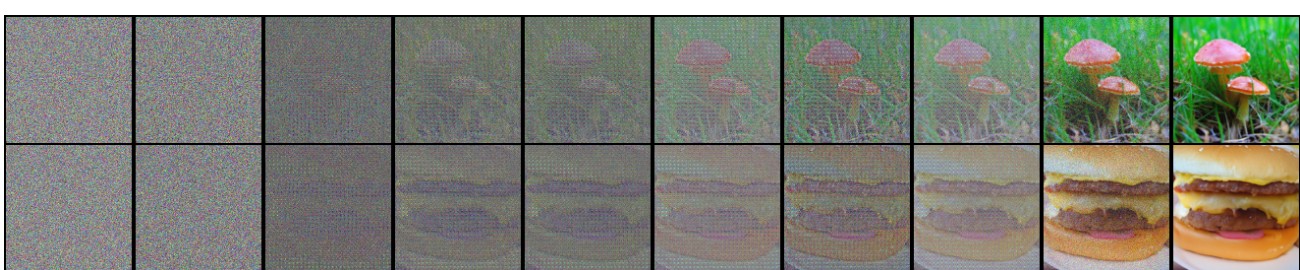

*Figure 8.* From left to right, the sampling trajectory from the model on ImageNet 128x128 with 8 flow blocks. The visualization includes the final denoising step.

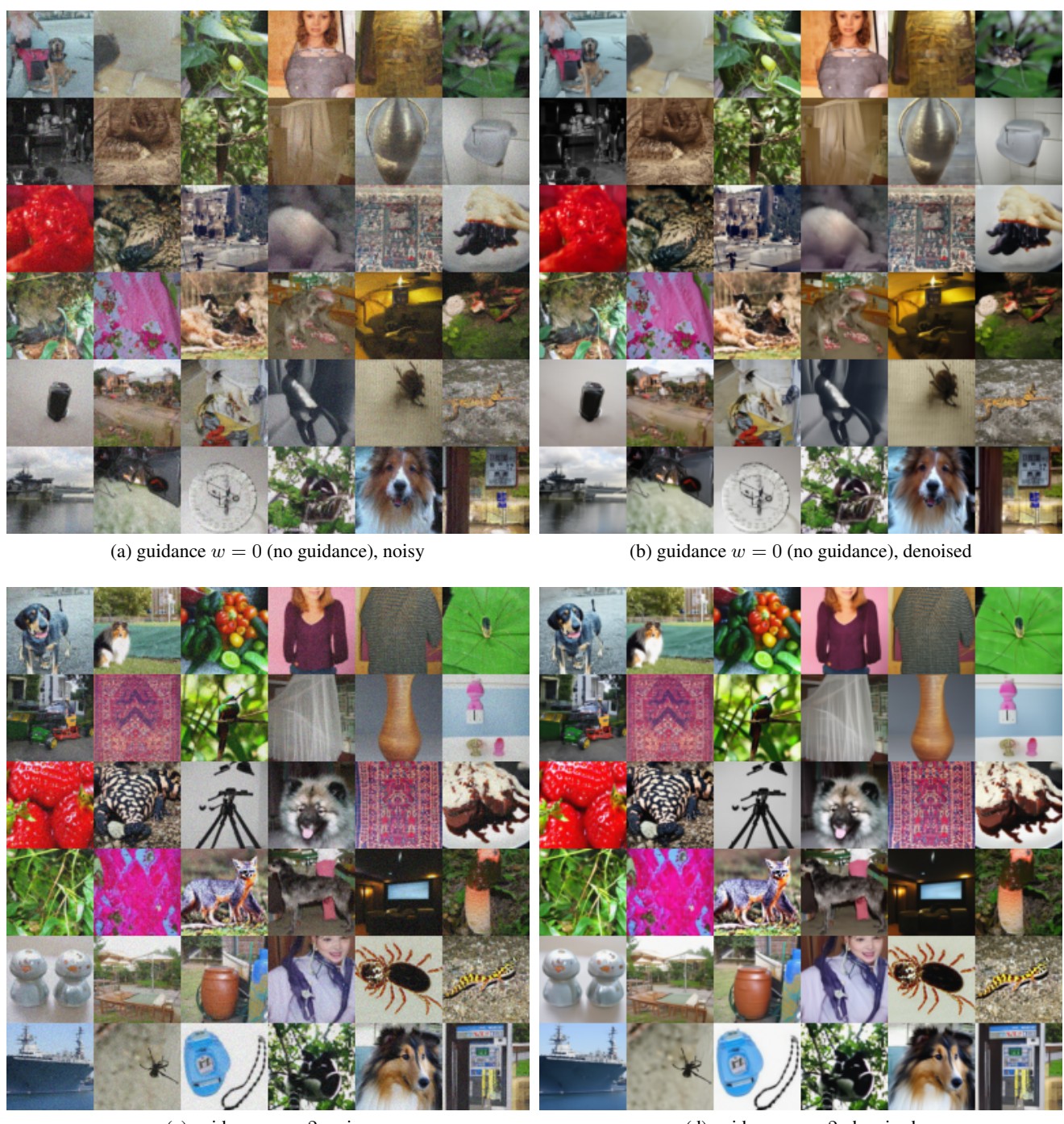

(a) guidance $w = 0$ (no guidance), noisy

(b) guidance $w = 0$ (no guidance), denoised

(c) guidance $w = 2$, noisy

(d) guidance $w = 2$, denoised

*Figure 9.* Uncurated samples with a fixed set of initial noise from the model trained on conditional ImageNet 64x64.

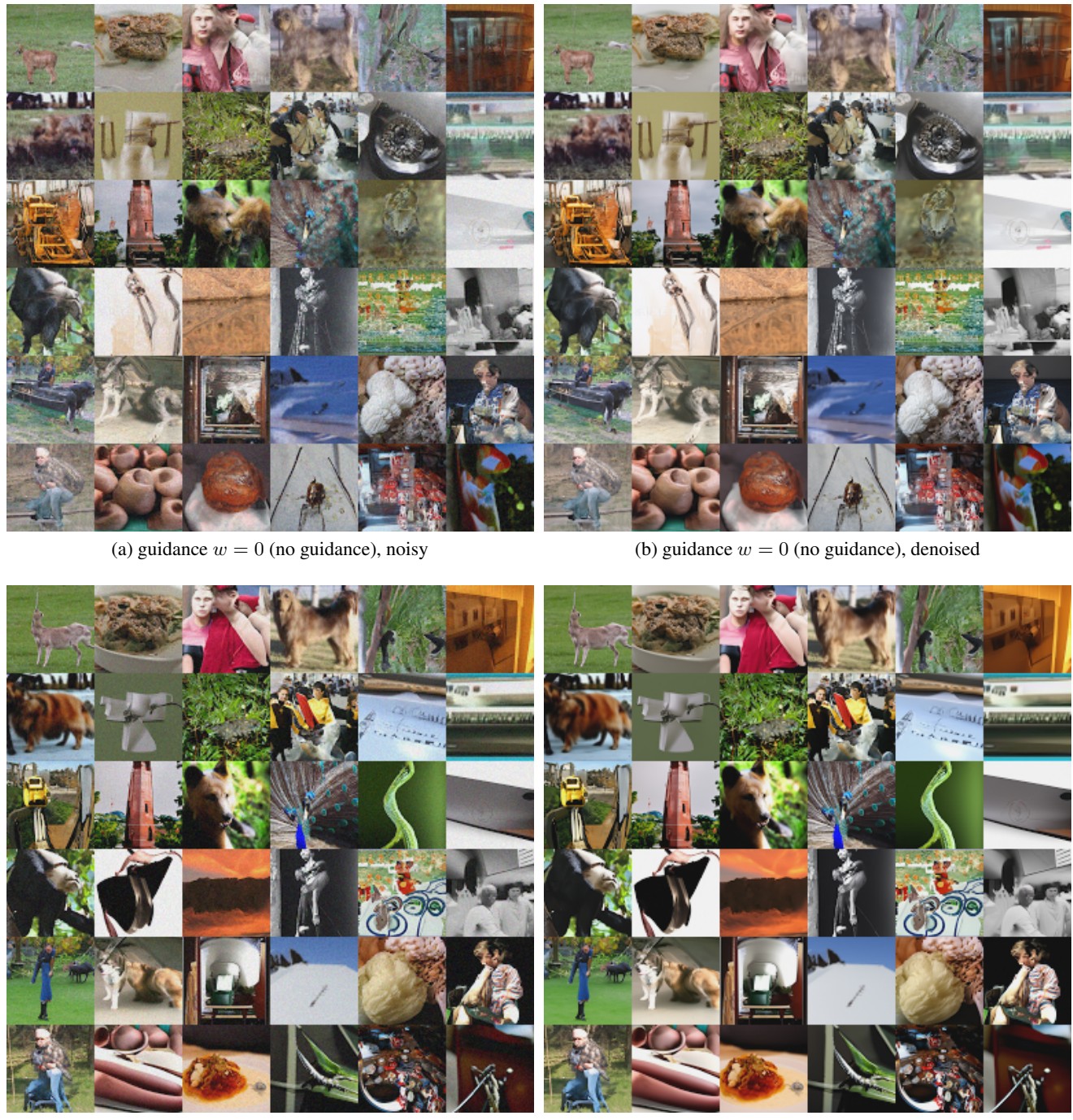

(a) guidance $w = 0$ (no guidance), noisy

(b) guidance $w = 0$ (no guidance), denoised

(c) guidance $w = 0.15, \tau = 0.2$, noisy

(d) guidance $w = 0.15, \tau = 0.2$, denoised

*Figure 10.* Uncurated samples with a fixed set of initial noise from the model trained on unconditional ImageNet 64x64.

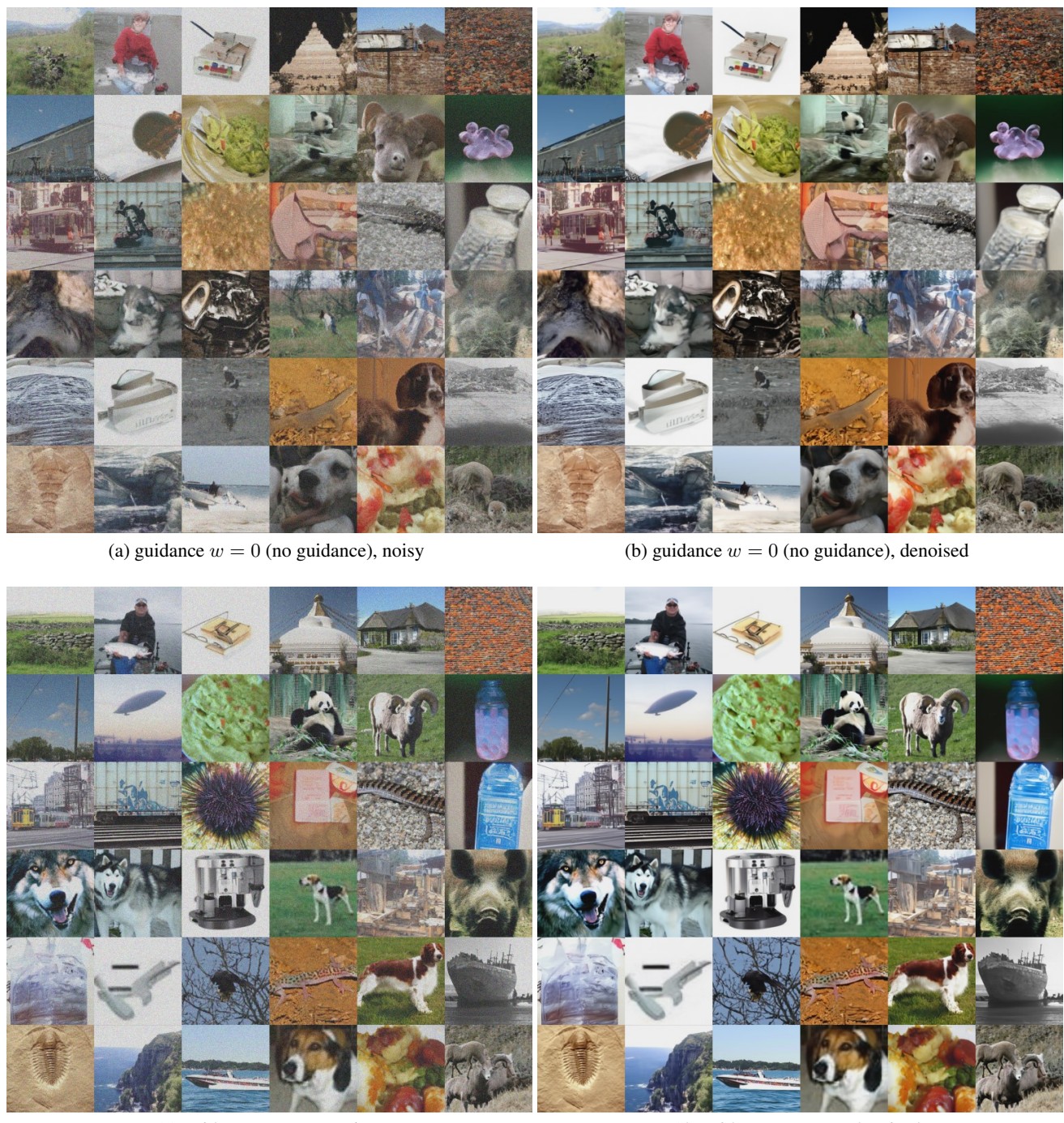

(a) guidance $w = 0$ (no guidance), noisy

(b) guidance $w = 0$ (no guidance), denoised

(c) guidance $w = 2.5$, noisy

(d) guidance $w = 2.5$, denoised

*Figure 11.* Uncurated samples with a fixed set of initial noise from the model trained on conditional ImageNet 128x128.

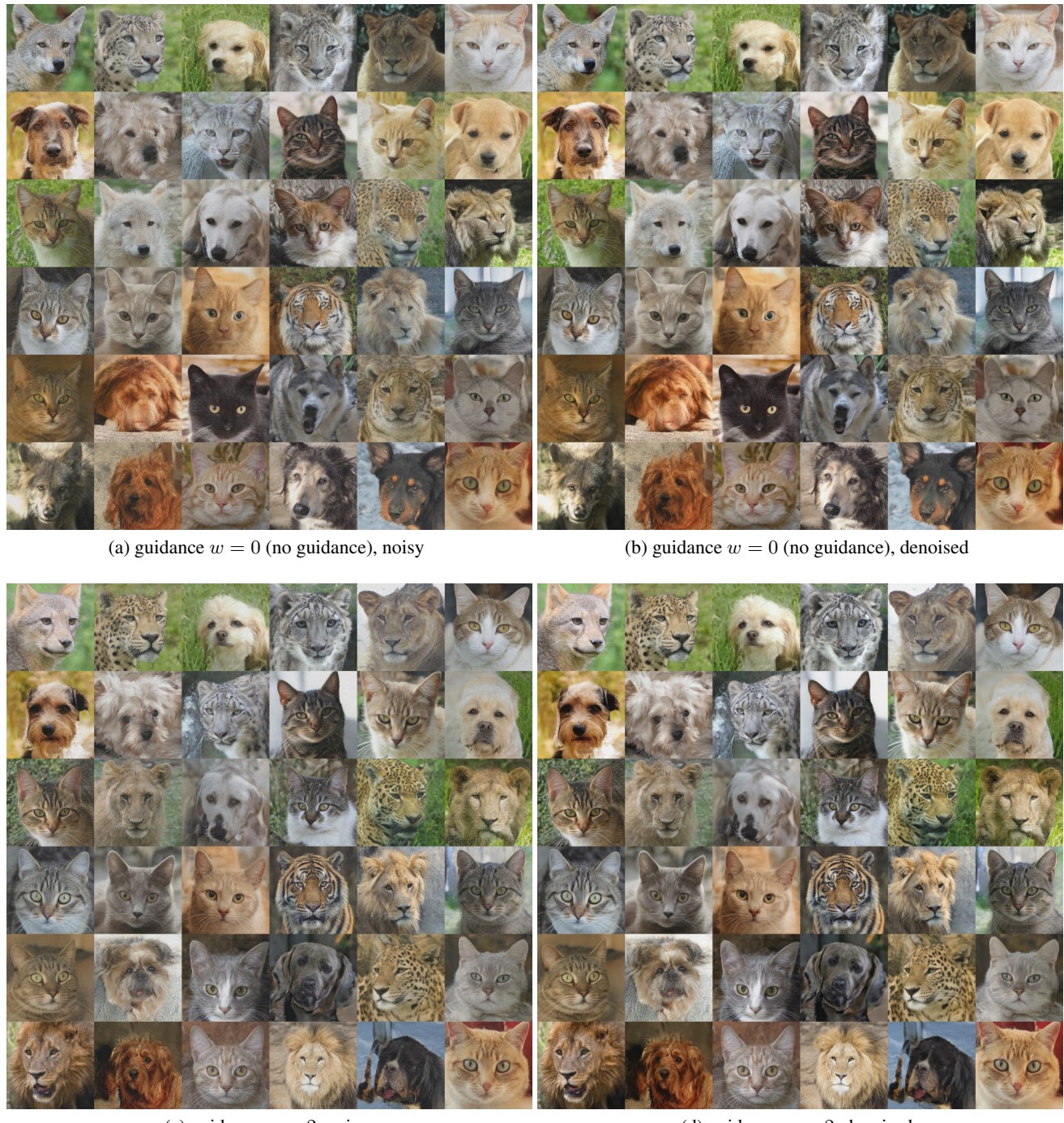

(a) guidance $w = 0$ (no guidance), noisy

(b) guidance $w = 0$ (no guidance), denoised

(c) guidance $w = 2$, noisy

(d) guidance $w = 2$, denoised

*Figure 12.* Uncurated samples with a fixed set of initial noise from the model trained on conditional AFHQ 256x256.

