# OpenReview forum: "Normalizing Flows are Capable Generative Models"
_ICML.cc/2025/Conference — ICML 2025 oral_

### Official Review · Reviewer_fuKH · 2025-02-22

**Overall Recommendation:** 3

**Summary:**

The paper proposes a new architecture called TarFlow, which is a Transformer-based variant of Masked Autoregressive Flows (MAFs). TarFlow achieves a high-performance normalizing flow (NF) model by stacking autoregressive Transformer blocks on image patches and alternating the autoregressive direction between layers. Additionally, the paper introduces three techniques to improve sample quality: Gaussian noise augmentation, a post-training score-based denoising technique, and efficient guidance recipes for both the class conditional and unconditional models.

**Claims And Evidence:**

I think the claims regarding the normalizing flow part is solid, but I'm not so persuaded in terms of motivation part, which claims that the normalizing flow have seen limited practical adoption. Shouldn't flow matching techniques has already been widely applied? Can author address the foundamental differences of the proposed TarFLow and current flow matching methods?

**Essential References Not Discussed:**

None

**Experimental Designs Or Analyses:**

1. As mentioned before, I think this paper lacks comparison and discussion with flow matching method, which should be closely related to the realm of normalizing flow.

2. From the quantitative experiments, the performance of the TarFlow is not comparable with diffusion-based method, can the author provide analysis regarding this phenomenon?

3. In generation quantitative experiments, it lacks comparison with other flow and autoregressive model, please add more baseline results.

**Methods And Evaluation Criteria:**

1. In the section that introduces guidance technique, I'm wondering how the class label conditioning is added in the transformer architecture, through input concatenation or cross-attention?

2. The tasks and benchmark datasets make sense, but I think the author should give a little bit introduction to the less commonly-used evaluation criteria like Bits per dim (BPD) evaluation in table 2.

**Other Comments Or Suggestions:**

None

**Other Strengths And Weaknesses:**

One subtle suggestion is that the top visualization part of figure 4 is not clear enough. It's hard to distinguish the difference between the raw samples and denoised samples.

**Questions For Authors:**

None

**Relation To Broader Scientific Literature:**

NA

**Theoretical Claims:**

This paper does not propose theoretical claims that needs mathematical derivation or proof. Its contribution mostly lies in the design of Transformer-based normalizing flow architecture.

---

> ### Author Rebuttal · Authors · 2025-04-01
>
> ## Q: Motivation and difference to flow matching
>
> A: We’d like to clarify that the exact notion of Normalizing Flow (NF) we consider is here fundamentally different from the modern notion Flow Matching (FM) method. In our paper, we follow the conventional notation of NFs that exclusively refer to a maximum likelihood method that uses the change of variable technique of probability [1, 2, 3]. One of their distinct properties is that the training loss can be **exactly computed** for each training example. Flow matching [4] on the other hand, denotes a training technique that can be considered a variant of diffusion models, see [5] for such illustration, which does not follow a maximum likelihood objective. Also, the training loss of FM relies on a **stochastic approximation** of an expectation over time steps, and over the noise within each step. Moreover, it was shown that FM’s training objective follows a modified variational lower bound of likelihood [8], which again is fundamentally different from NFs's exact likelihood nature. In fact, this difference is also documented in [4], see the first paragraph of Section 5. The confusion may stem from the reference to continuous normalizing flow (CNF) in [4], which considers a **generalized notion of NFs** which deterministically map noise to data, regardless their training objectives. However, please note that the CNF here is mathematically equivalent to the probability flow [7], or ODE/DDIM style inference path in diffusion models [6]. This is also the reason why we have categorized all diffusion model and flow matching results under the term Diff/FM in our comparisons, see eg Table 2.
>
> We hope this clarifies the difference, and we will be happy to make this more explicit in our paper, constraining the scope of our discussion of NFs to methods that directly follow the MLE objective with the change of variable formula.
>
> ## Q: Label conditioning implementation
>
> A: We adopt a simple implementation of label conditioning, where we add the label embedding directly to the position embeddings in each flow block. During guidance, the unconditional predictions are obtained by averaging the label embeddings of all classes and adding it to the position embeddings.
>
>
> ## Q: Introduction to BPD
>
> A: Thank you for the suggestion, we are happy to provide more context on it. In simple terms, BPD measures the average log probability for all the pixels of an image, where the probability is computed as a discrete one among 256 possible pixel values.
>
>
> ## Q: Results worse than diffusion models
>
> A: We agree that the current results of TarFlow is still lacking behind the best tuned diffusion models. We argue that for any new type of method, it usually takes time for it to be collectively improved by a community before it reaches the SoTA performance, and this was definitely the case for diffusion models too. We look forward to working with the generative modeling community together to further improve the upper bound of normalizing flow methods.
>
>
> ## Q:  Comparison with other flow and autoregressive model
>
> A: Our comparisons have been focusing on mainstream continuous modeling method, and we have identified GANs, diffusion/flow matching models as the representative categories. Also note that we did not include traditional NF baselines in the FID comparisons as they were generally under performing and we were not able find comparable FID results reported in the literature. Following the reviewer's suggestion, we will also include representative baselines from Autoregressive models. Also, we would be happy to include more baselines for comparison if the reviewer is able to provide more specific references.
>
> ## Q: Figure 4 not clear
>
> A:  We apologize for the visualization clarity, but the main cause for it is that the noise we apply is indeed very small and it does require a closer look to recognize the effect of denoising. We refer the review to more visualizations in the supplementary material, ie Figure 9-12, which should allow one to observe the effect of noise more clearly.
>
> # References
> [1] Density estimation by dual ascent of the log-likelihood, Tabak et al, Communications in Mathematical Sciences, 2010
>
> [2] Variational inference with normalizing flows, Rezende & Mohamed, ICML 2015
>
> [3] Nice: Non-linear independent components estimation, Dinh et al, ICLR 2014
>
> [4] Flow Matching for Generative Modeling, Lipman et al, ICLR 2023
>
> [5] Diffusion Meets Flow Matching: Two Sides of the Same Coin, Gao et al, ICLR Blogposts 2025
>
> [6] Denoising Diffusion Implicit Models, Song et al, ICLR 2021
>
> [7] Score-Based Generative Modeling through Stochastic Differential Equations, Song et al, ICLR 2021
>
> [8] Understanding Diffusion Objectives as the ELBO with Simple Data Augmentation, Kingma & Gao, NeurIPS 2023

---

### Official Review · Reviewer_EWAZ · 2025-03-03

**Overall Recommendation:** 5

**Summary:**

The paper presents a normalizing flow architecture and training pipeline for image generation that significantly improves previous normalizing flow models and obtains competitive performance when compared with diffusion models and GANs. The architecture uses a masked transformer backbone to implement RealVPN-type partitioned layers. Training is performed on noised images with a fixed level of noise, which is then removed at inference time using Tweedie’s formula, in what essentially is a single step of denoising diffusion. Importantly, this is done using the existing network without additional training. The paper also introduces an interesting guidance scheme inspired by classifier-free guidance but at the level of the attention layers.

**Claims And Evidence:**

The claims are convincing and well supported by the experiments, with several ablation studies elucidating the relative contributions of the novel ideas.
For reasons explained below, I do not think that the exceptional likelihood performance is particularly interesting, since the data approximately lives in a lower-dimensional manifold.

**Essential References Not Discussed:**

The literature coverage concenring normalizing flows, diffusion models and tautoregressive models is appropriate.

**Experimental Designs Or Analyses:**

The experiments follow standard image generation and evaluation approaches widely used in the image generation literature.

**Methods And Evaluation Criteria:**

The methods and evaluation metrics are in line with the modern literature. Several ideas from transformers and diffusion models are integrated elegantly in the normalizing flow framework. The authors show a good mastery of the modern generative modeling landscape.

**Other Comments Or Suggestions:**

None

**Other Strengths And Weaknesses:**

Strengths:
It is very nice to see modern work on normalizing flows, and I think that the authors did a great job in upgrading these models to near the current SOTA using both ideas from transformers and diffusion models.

The main strengths of the paper are:

The proposed architecture is both elegant and powerful and I think it could offer a starting point for a renaissance in normalizing-flow research even outside computer vision.
The guidance scheme, if original, is very interesting and it could be applied to all sorts of transformer architectures.
While it is somewhat disappointing that the model needs to work on noised data, I appreciated the elegance of the denoising solution inspired by diffusion theory. I really appreciate how the authors are borrowing elegant ideas from different approaches while integrating them elegantly within the normalizing flow framework.
Weaknesses:
While the results are convincing, the paper is also a clear indication that normalizing flows, even with very modern components, fail when trained to generate noiseless images.
 I think that this simply reflects a fundamental problem with likelihood-based models such as NFs when trained on data such as images that are supported on a lower dimensional manifold-like structure. The issue is that the likelihood diverges if the model approaches the correct support of the data, regardless of how well it is fitting the distribution restricted to the manifold. This phenomenon is known as ‘manifold overfitting’ [CITE]. In the case of NFs on noiseless images, the optimum of the loss is situated near or at a singular point of non-invertibility, which leads to unstable training. So said, this is not really a weakness of the paper, which does a good job at characterizing this behavior empirically.

Due to manifold overfitting and divergence of the likelihood, I do not think that the likelihood values are of particular interest, since on manifolds a bad generative model can reach arbitrarily high likelihood simply by fitting some section of the support of the data. In fact, in this case we see exceptionally high likelihood values together with FIDs that are good but not exceptional. Again, I do not consider this as a real weakness of the paper, but I do not put much weight in the exceptionally good BPD numbers.

**Questions For Authors:**

I am really intrigued by your guidance method. Are you sure that it is not currently used in the LLM literature? I do not know the literature well enough to know. If not, it would be very interesting to use it in more conventional transformers as well.

**Relation To Broader Scientific Literature:**

The paper offers an elegant integration of several modern generative modeling ideas"
1) It combines the general RealNVP framework with a masked transformer architecture.
2) It uses a guidance scheme inspired by classifier-free guidance in diffusion models. However, here the guidance is used within the different layers opf the normalizing flow architecture.
3) It uses a denoising technique that uses the basic formulas of generative diffusion models.

**Theoretical Claims:**

The paper does not make precise theoretical claims. The formulas and derivations are solid.

---

> ### Author Rebuttal · Authors · 2025-04-01
>
> We thank the reviewer for acknowledging our contributions and we agree most of your assessments.
>
> ## Q:  Limitations of likelihood-based models
>
> A: An interesting point of discussion the reviewer brought up is the fundamental limitations of likelihood-based models. We agree with the reviewer that likelihood method, especially density estimation such as NFs, can be ill behaved on real data that’s distributed on a narrow manifold. And we also agree that TarFlow needing input noise resonates with this view, which put in other words, suggests that density estimation is indeed a different task than generative modeling. We also believe that this observation is consistent with the findings in diffusion models, whereas the reweighting the likelihood based loss function is beneficial for sampling quality [1, 2].
>
>
> ## Q: Significance of the BPD benchmark
>
> A: A related question is whether one should value the strong BPD results that TarFlow achieves. We believe so, based on a subtle yet interesting point detailed below. Note that the BPD metric evaluates the discrete probability of quantized pixels, rather than the continuous density of real pixel values. This is a key difference because, unlike continuous density, discrete probability can always be faithfully evaluated even when the input has much lower intrinsic dimension (eg, concatenating a constant dimension to a discrete variable multiplies the joint discrete probability by 1, instead of infinity). When using a density estimation method to evaluate BPD, one would first inflate the discrete values $\tilde{x}$ to a local hyper cube $C(\tilde{x})$ that is disjoint from other discrete points (which is the case for the dequantization uniform noise), and convert the density model to discrete probability via integration, as in $\tilde{p}(\tilde{x}) = \int_{x \in C(\tilde{x})} p_{model}(x)dx$ (this is alo explained in the first paragraph of Section 2.4). Due to the same reasoning, the discrete probability result (hence BPD) is comparable among difference families of models, including discrete modeling methods using autoregressive models and continuous ones such as diffusion and NFs. In addition, the BPD metric has a concrete grounding itself, which essentially translates to the theoretical lower bounds for lossless compression of the target distribution [3], which is another way to prove that BPD does not degenerate. Therefore, we believe that TarFlow achieving SoTA BPD among different modeling methods is a strong indication of its raw modeling capacity.
>
>
> ## Q: temperature based guidance for LLMs
>
> A: We'd like first to confirm that we did come up with the attention based temperature guidance on our own, and we are not aware of a similar idea being deployed in other context such as LLMs. Like the reviewer, we are also intrigued by its generality and we believe that it does have potential to be applied to all Transformer based generative models as well. For LLMs, specifically, we speculate that guidance had received relatively little exploration likely due to the prevalence of finetuning, and it would indeed be interesting future work to further investigate its compatibility with our temperature based guidance.
>
> # References
> [1] Denoising Diffusion Probabilistic Models, Ho et al, NeurIPS 2020
>
> [2] Variational Diffusion Models, Kingma et al, NeurIPS 2021
>
> [3] IDF++: Analyzing and Improving Integer Discrete Flows for Lossless Compression, Berg et al, ICLR 2021

---

### Official Review · Reviewer_jjSB · 2025-03-08

**Overall Recommendation:** 3

**Summary:**

This paper proposes to integrate visual transformer architecture into Real NVP Normalizing Flows. Over the past years normalizing flows has been inferior to other types of generative models; particularly when compared to diffusion models. This paper claims that the reason for that is the design limitation of normalizing flows, therefore by using transformer backbones with causal attention mechanisms they are able to improve the generation performance and enhance the scalability of the model.

The approach shares similarity with Masked Autoregressive Flows (MAF), where it is extended to patches rather than pixels. Additionally, the input of the model is injected with small noise for smoothing (in probability terms) the discrete data distribution. Which in turn, requires an extra final denoising step, performed using a score based model.

The paper also presents a guidance mechanism, similar to the one used in diffusion models. Classifier free guidance in particular. Enabling the generation to be controlled externally.

**Claims And Evidence:**

- The paper claims and shows empirically that Normalizing Flows indeed need a more strong deep network architecture to be able to compete with state-of-the-art diffusion models.

The paper claims that the approach is scalable, similar to what has been done in diffusion models, however:

- The approach is not tested on ImageNet $256\times 256$, it is only trained on AFHQ at this resolution, which is a relatively homogeneous dataset, and there aren't any quantitative metrics that support this claim.

- Comparing the reported results at $128\times 128$ actually contradicts with the scalability claim. In contrast to $64\times 64$, where the FID score is inline with the diffusion scores, the result on $128\times 128$ is actually worse.

**Essential References Not Discussed:**

Although it is mentioned in the related work section, but it would be much better to discuss the main differences between this method and the one proposed in Patacchiola et al., 2024.

**Experimental Designs Or Analyses:**

The design scheme is straightforward, the paper uses transformer based architecture to predict the coupling of the bijection in an auto-regressive manner.

**Methods And Evaluation Criteria:**

Strengths:
- The approach is evaluated properly on different datasets.

- The results shown in the paper are promising, competitive with state-of-the-art diffusion models.

Weaknesses:

- The paper is missing an ablation study on the effect of the denoiser choice.

- The results without the noise injection are not shown in the paper, especially that the paper emphasizes the importance of it. I think it is very important to highlight the drawbacks of not adding the noise more clearly (diversity, mode collapse, textures, etc.).

- The paper shows the results only for causal transformer network, I think there should be an additional ablation study that demonstrate why this choice is preferable over non-causal transformers.

**Other Comments Or Suggestions:**

- Section 2.5 can be reduced, there is no need to provide redundant details about Tweedie’s relation of score and denoising if your target is just to denoise the final image.

**Other Strengths And Weaknesses:**

Other strengths:

- The paper is clear and well written.

Other weaknesses:

- The results without the noise injection are not shown in the paper, especially that the paper emphasizes the importance of it. I think it is very important to highlight the drawbacks of not adding the noise more clearly (diversity, mode collapse, textures, etc.).

- Obtaining high-fidelity generation results requires using a denoising network for post-processing the output.

- The novelty of the approach is relatively limited, particularly compared to Patacchiola et al., 2024.


- Normalizing Flows other than Real NVP are not examined.

**Questions For Authors:**

- What is the denoiser architecture used in the final stage?

- What is the overhead and added computational cost of using a denoising network?

- How does not adding noise at all affects the results? in terms of diversity, mode collapse, and quantitatively (FID).

 - In terms of computations, how many

- How important is the causality part of the method? Does the method also work if the transformer was not causal? How does that affect the results?

**Relation To Broader Scientific Literature:**

The paper highlights the advantage of using capable architectures, transformer networks in particular,  in Normalizing Flows. Prior flow based models struggled to compete with state-of-the-art diffusion models, and this work offers a very competitive alternative (for lower resolutions at least).

**Theoretical Claims:**

N/A

---

> ### Author Rebuttal · Authors · 2025-04-01
>
> We thank the reviewer for the detailed review, and please see our responses below.
>
> ## Q:  Scalability claim
>
> A: Our claim about the scalability of TarFlow is within the context of different model sizes and training FLOPs on a given dataset, which is supported by evidence in Sec 3.5 and Figure 6. Moreover, for the comparison of ImageNet 128x128 vs 64x64, we do not agree that the former is worse. Note that the absolute FID numbers are not comparable across datasets, but it is pretty clear that our ImageNet 128x128 samples are visually better than those on 64x64 (Figure 11 vs Figure 9).
>
> In addition, we have conducted further experiments on ImageNet 256x256 with a model of ~1.4B parameters, and we are able to achieve very competitive results, with a 50K sample FID of 4.00 on pixel space. This number again places TarFlow in a diffusion dominated regime, see the table below as a comparison.
>
> | Method (modeling pixels) | ImageNet 256x256 FID 50K |
> | --------  | ------- |
> | Simple Diffusion | 3.75|
> | ADM-G | 4.59 |
> | RIN | 4.51 |
> | TarFlow | 4.00 |
>
> ## Q: Denoising & Section 2.5
> A: We believe that the reviewer has a serious misunderstanding of our contributions. To clarify, denoising is performed with the same TarFlow model we train, and **we do not use a separate denoising network**. The exact recipe is explained in Equation 8, where $\log p_{model}(y)$ corresponds to the negative training loss of the TarFlow model. This also explains why Section 2.5 is a critical piece of our method, as it is not previously clear that NF models empirically give rise to accurate score estimates, making it a suitable choice for denoising in line with the Tweedie’s formula. Note that this point is also correctly recognized and acknowledged by Reviewer EWAZ as a strength of our paper.
>
> As for speed, the denoising step adds a minimum overhead, due to its parallel nature. For actual measurements, we benchmarked the model we used for ablations, ie, the one from Figure 4, it takes 13.5 seconds to generate a noisy batch of examples, and 0.14s to denoise them.
>
> ## Q: Results without noise injection
> A: We assume that the reviewer is interested in the dequantization uniform noise setting, which is a standard in the NF literature [1] — and also the simplest way to make modeling pixels a valid density estimation problem (see, eg, Sec 3.1 of [3] for a discussion on this). As noted in the first paragraph of Section 3.3, this setting has poor numerical stability which makes it difficult to compare with other settings fairly. Nonetheless, during the rebuttal period we managed to train such a model by using fp32 instead of our default bf16, and performed 50K FID evaluation by skipping many samples with NaNs. See results in the table below.
>
> ## Q: Non-causal transformer
> A: First, the causal architecture is a necessary component for implementing the AR flow. More specifically, it allows us to compute the determinant analytically, and also invert the transformation explicitly, due to the Jacobian of the causal transformation being lower triangular. Both conditions will break if one directly applies a non-causal Transformer. That being said, the closest baseline we can think of that’s using a non-causal Transformer is to follow the channel coupling design from [1].  We performed such an ablation, and the result is shown below.
>
> ## Q: Normalizing Flows other than Real NVP
> A: We have also trained a volume preserving version, by enforcing the logdet terms to be zero. See results below.
>
> ### Summary of additional ablations, where the baseline experimental setting follows that of Figure 4. All variants have inferior results than the default TarFlow setting.
>
> | Variant   | ImageNet 64x64 FID 50K, cfg = 0 | ImageNet 64x64 FID 50K, cfg = 2|
> | --------  | ------- | ------- |
> | TarFlow default |25.3| 5.7 |
> | dequantization uniform noise | 43.6 | 21.9 |
> | non-causal architecture with channel coupling | 50.3 | 20.4 |
> | volume preserving   | 81.5 | 51.0 |
>
> ## Q: Novelty
> A: We believe that TarFlow has significant novelty compared to [2]. The similarity is that they both apply Transformers to MAF. However, it is important to note that [2] does not provide a scalable recipe for high dimensional inputs. In particular, [2] only considers models with a single flow, whereas TarFlow stacks multiple flows with alternating directions, and show that this can be trained well. As shown in Figure 6(b) of our paper, using one flow (T=1) significantly limits the model’s capacity and it becomes a degenerated model for images. The other obvious difference lies in our usage of noise augmented training, denoising and guidance. All these critical pieces are completely missing in [2].
>
> # References
> [1] Density estimation using Real NVP, Dinh et al, ICLR 2017
>
> [2] Transformer Neural Autoregressive Flows, Patacchiola et al., 2024
>
> [3] Flow++: Improving Flow-Based Generative Models with Variational Dequantization and Architecture Design, Ho et al, ICML 2019

---

### Official Review · Reviewer_M2FB · 2025-03-14

**Overall Recommendation:** 4

**Summary:**

This paper scales up masked autoregressive flow (MAF) with powerful transformer architecture along with several techniques such as classifier guidance, noise augmentation and achieve good performance on many datasets including high resolution AFHQ and multimodal dataset Imagenet.

## update after rebuttal
The rebuttal has resolved my concerns. After careful consideration, I decided to vote Accept for this paper.

**Claims And Evidence:**

Yes, paper provides clear representation and convincing evidence to all claims

**Essential References Not Discussed:**

No, I find the reference is sufficient

**Experimental Designs Or Analyses:**

The experimental design is well-conducted and support all the paper's claim.

**Methods And Evaluation Criteria:**

Yes, the methods and evaluation criteria are standard for generative model.

**Other Comments Or Suggestions:**

No

**Other Strengths And Weaknesses:**

**Strength**

The paper presentation is clear and easy to understand.

The experiment shows promising potential for normalizing flow to scale up, in comparison to diffusion, autoregressive and GAN models

The ablation is fully provided

Several proposed techniques such as noisy augmentation, transformer backbone for MAF and cfg are first-time investigated in MAF

**Weakness**:


The sampling time is very slow compared to standard autoregressive models.

The architecture of model is quite heavy 8x8 = 64 transformer blocks, which is about 2 time larger than DiT XL

The method is not new and a little bit limited novelty since classifier free guidance and data augmentation are existed techniques

**Questions For Authors:**

Is the anyway to develop faster inference model based on TarFlow ?

**Relation To Broader Scientific Literature:**

This paper scale up MAF, an existing technique to train normalizing flow to show the scalability of normalizing flow class of generative models. The paper's result is quite interesting and promising.

**Theoretical Claims:**

There is no theoretical claim in this paper.

---

> ### Author Rebuttal · Authors · 2025-04-01
>
> We thank the reviewer for acknowledging our contributions and we answer each of the questions below.
>
> First of all, please note that we do have a supplementary material where we include more experimental settings, related work and results which we believe might be interesting to the reviewer.
>
>
> ## Q: The sampling time is very slow compared to standard autoregressive models.
>
> A:  Given a sequence length N (ie, number of patches), number of AR flows T and the Transformer depth for each flow L, the inference cost of TarFlow evaluates to $O(TLN^2)$. This cost will be equivalent to that of an autoregressive model on the same sequence but with a depth of $TL$. Therefore, the inference speed comparison between TarFlow and an AR model boils down to the combined depth, and the two families of models actually have the same inference time complexity given the same depth budget.
>
>
> ## Q: The architecture of model is quite heavy 8x8 = 64 transformer blocks, which is about 2 time larger than DiT XL
>
> A: Related to the previous question, our design philosophy has been to train strong models and prove that NFs are capable learning objectives, whereas we did not emphasize on reducing the total depth. As for the comparison to DiT, we argue that they are also not directly comparable, due to two reasons. 1. DiT conducts experiments in the latent setting, which is known to simplify the modeling difficulty, whereas TarFlows are pixel based models. 2. The inference mode of DiTs is actually much deeper than TarFlow, due to the need of calling multiple NFEs.
>
>
> ## Q: The method is not new and a little bit limited novelty since classifier free guidance and data augmentation are existed techniques
>
> A: We respectfully disagree with the reviewer’s assessment on novelty. It is indeed true that all the essential ingredients in this work has been previously established techniques, this includes MAF, Transformers, noise augmentation, denoising and guidance. However, in the context of normalizing flows, the exact recipe of using them has been unknown, and as the reviewer already acknowledged, we are the first work to show the full potential of correctly combining them together. The fact that these individual techniques are standard only strengthens the simplicity aspect of our work, and it should not be treated as a penalty for novelty. In fact, the same argument can be made for many groundbreaking works — for example, denoising autoencoders are well studied subjects [1], but correctly applying it to generative modeling [2, 3] is still highly novel and it gives rise to the diffusion and score based generative model revolution.
>
>
> ## Q: Is the anyway to develop faster inference model based on TarFlow ?
>
> A:  Although we did not focus on inference speed in this work, we are excited by several possibilities of improving it in the future work. First of all, there is a large body of literature (eg, speculative decoding [4]) dedicated to speeding up the inference of autoregressive models, which TarFlow can already benefit from, due to its autoregressive architecture. Another promising direction is distillation, which has achieved great success in diffusion models [5]. And we believe that distillation should be naturally compatible with TarFlow, and arguably more so than with diffusion models, as we have an explicit bijective mapping between an input x and noise z.
>
> # References
> [1] Extracting and composing robust features with denoising autoencoders, Vincent et al, ICML 2008
>
> [2] Denoising Diffusion Probabilistic Models, Ho et al, NeurIPS 2020
>
> [3] Generative Modeling by Estimating Gradients of the Data Distribution, Song & Ermon, NeurIPS 2019
>
> [4] Fast inference from transformers via speculative decoding, Leviathan et al, ICML 2023
>
> [5] Progressive Distillation for Fast Sampling of Diffusion Models, Salimans & Ho, ICLR 2023

---

### Decision · Program_Chairs · 2025-05-01

**Decision:**

Accept (oral)

**Comment:**

This paper has solid consensus for acceptance. It tackles an interesting problem of making normalizing flows truly performant models, rather than just density estimators.